# SGORNN: Combining Scalar Gates and Orthogonal Constraints in Recurrent Networks

## Abstract

Recurrent Neural Network (RNN) models have been applied in different domains, producing high accuracies on time-dependent data. However, RNNs have long suffered from exploding gradients during training, mainly due to their recurrent process. In this context, we propose a variant of the scalar gated FastRNN architecture, called Scalar Gated Orthogonal Recurrent Neural Networks (SGORNN). SGORNN utilizes orthogonal linear transformations at the recurrent step. In our experiments, SGORNN forms its recurrent weights through a strategy inspired by Volume Preserving RNNs (VPRNN), though our architecture allows the use of any orthogonal constraint mechanism. We present a simple constraint on the scalar gates of SGORNN, which is easily enforced at training time to provide a theoretical generalization ability for SGORNN similar to that of FastRNN. Next, we provide bounds on the gradients of SGORNN, to show the impossibility of (exponentially) exploding gradients. Our experimental results on the addition problem confirm that our combination of orthogonal and scalar gated RNNs are able to outperform both predecessor models on long sequences using only a single RNN cell, and outperforms LSTM. We further evaluate SGORNN on the HAR-2 classification task, where it improves upon the accuracy of FastRNN, VPRNN, and LSTM using far fewer parameters than FastRNN or LSTM. Finally, we evaluate SGORNN on the Penn Treebank word-level language modelling task, where it again outperforms its predecessor architectures and shows comparable performance to LSTM using far less parameters. Overall, this architecture shows higher representation capacity than VPRNN, suffers from less overfitting than the other two models in our experiments, benefits from a decrease in parameter count, and alleviates exploding gradients when compared with FastRNN on the addition problem.

## 1 Introduction

Recurrent Neural Networks (RNN) are models for sequence processing that have found applications in many areas, such as statistical machine translation (Cho et al., 2014), credit score calculation (Babaev et al., 2019), vehicle mobility prediction (Liu & Shoji, 2019), and music generation (Agarwala et al., 2017). Despite their wide applicability, RNNs have long suffered from exploding gradients, mainly due to the repeated application of the network's hidden weights matrix during gradient computation (Hochreiter & Schmidhuber, 1997). Several strategies have been proposed to solve this problem, including models with orthogonal/unitary constraints on the hidden weights (Taylor-Melanson et al., 2021; Casado & Martínez-Rubio, 2019; Zhang et al., 2018; Vorontsov et al., 2017; Arjovsky et al., 2016; Jing et al., 2016). Such orthogonal models usually have fewer parameters than standard RNN architectures, which reduces the representation capacity and leads to less accurate models. Specifically, Taylor-Melanson et al. (2021) noted that a stack of two RNN cells is required to accurately model the addition problem for RNNs using the orthogonal VPRNN architecture when sequences become very long, and Rusch & Mishra (2020) noted that the EXPRNN architecture (Casado & Martínez-Rubio, 2019) fails to converge on the addition problem for very long sequences. To combat the problem of reduced representation capacity, Jing et al. (2019) proposed an LSTM-based architecture called the gated orthogonal recurrent unit (GORU)

which was experimented with using the EURNN orthogonal parametrization (Jing et al., 2016). While the GORU architecture is a very useful and novel orthogonal RNN construction, it adds many parameters when compared to typical orthogonal RNNs, and thus may not be suited for embedded or mobile applications.

Recently, Kusupati et al. (2019) proposed the FastRNN architecture, which adds a residual connection to a standard recurrent network using a scalar gating mechanism. FastRNN outperforms several well-known unitary RNN models on many classification tasks. These scalar gated RNN models have a generalization error gap which scales with the product of the length of the sequences being processed and the non-residual scalar gate, which is established through constraints introduced on the scalar gates of the network that are generally not enforced while training. Because of this, the assumptions required to prove the generalization of FastRNN cannot be guaranteed to be satisfied in practice under the original formulation of the architecture.

Although the VPRNN architecture and many other RNNs alleviate exploding gradients by utilizing orthogonal constraints, the problem of increasing representation capacity for orthogonal RNNs without greatly increasing parameter counts still remains. Moreover, even with the benefit from the weighted residual connection, FastRNN can still suffer from exploding gradients when applied to long sequences as shown in Section 3.1, and Rusch & Mishra (2020) noted that FastRNN fails to converge when training on long sequences for the RNN addition problem. In order to solve these problems, we propose the Scalar Gated Orthogonal Recurrent Neural Network (SGORNN), which is a variant of FastRNN that uses orthogonal linear transformations at the recurrent step.

The constraints on the scalar gates of FastRNN required for generalization are dependent on bounds of the Frobenius norm of the hidden weights matrix, and of the sequence length of the data being processed. Both of these quantities may not be well-defined in practice, making it difficult to ensure generalization. In contrast with FastRNN constraints, we present a constraint that relies only on the scalar gates of the SGORNN model, which is feasible to be employed in practice and results in a generalization gap with similar growth to that of FastRNN. Our theoretical results also confirm that SGORNN is immune to the (exponentially) exploding gradients so often seen when training RNN models.

In summary, SGORNN can ensure its theoretical generalization by applying the constraints we provide, which also alleviates exploding gradients. Our addition problem experiment shows that SGORNN converges for long sequences with a single RNN cell, outperforming both predecessor architectures despite having far fewer parameters than FastRNN. Using real-world data, SGORNN shows superior generalization to its predecessor models and comparable performance to LSTM models with far more parameters.

The remaining sections of this paper are organized as follows. In Section 2, we describe the proposed SGORNN architecture, followed by the theoretical formulation on generalization and exploding gradients. Experiments are presented and discussed in Section 3. In Section 4, we present the conclusions obtained from the experiments and formulation, as well as proposed future work. Finally, Appendix A contains detailed information on the mathematical formulation used to provide the generalization bound for SGORNN, as well as the alleviation of exploding gradients.

## 2 SGORNN

The proposed Scalar Gated Orthogonal Recurrent Neural Network (SGORNN) uses orthogonal linear transformations at the recurrent step. Given an input sequence $\{\mathbf{x}_t\}_{t=1}^T \subset \mathbb{R}^n$ of length $T > 0$, we define the hidden states $\mathbf{h}_t \in \mathbb{R}^h$ of a SGORNN at time $t$ as:

$$\mathbf{h}_t = \alpha\sigma(\mathbf{W}\mathbf{x}_t + \phi(\Theta)\mathbf{h}_{t-1} + \mathbf{b}) + \beta\mathbf{h}_{t-1}. \tag{1}$$

where $\Theta$ holds the recurrent parameters of the architecture, $\phi$ is a mapping from recurrent parameters into the orthogonal matrices, $\mathbf{W} \in \mathbb{R}^{h \times n}$ and $\mathbf{U} = \phi(\Theta) \in \mathbb{R}^{h \times h}$ are matrices, $\mathbf{b} \in \mathbb{R}^h$ is a bias vector, and $\sigma$ is a nonlinear activation function, such as rectified linear unit (ReLU). We use the notation $\mathbf{U} = \phi(\Theta)$ for consistency with Kusupati et al. (2019) during analysis. As with FastRNN, the parameters $\alpha, \beta \in (0, 1)$ are set by the sigmoid function. For simplicity, we assume $\mathbf{h}_0$ is a vector of zeros, though, in theory, it can be made to be a trainable value as well.

For the sake of experimentation, we choose $\phi(\Theta)$ to be defined as the hidden weights matrix of the VPRNN architecture (Taylor-Melanson et al., 2021), a recently developed orthogonal RNN which is known to outperform several other unitary/orthogonal RNNs on classification tasks. It is worth mentioning that, although we use the VPRNN mapping in our experiments with SGORNN, an alternative orthogonal constraint such as that of EXPRNN (Casado & Martínez-Rubio, 2019) (with $\phi = \exp$) can also be applied to the proposed architecture. Following from Taylor-Melanson et al. (2021), the recurrent matrix is defined as:

$$\phi(\Theta) = \mathbf{R}_1 \mathbf{Q}_1 \ldots \mathbf{R}_{k-1} \mathbf{Q}_{k-1} \mathbf{R}_k \mathbf{Q}_k, \tag{2}$$

which is a case of SGORNN with $\text{dom}(\phi) = \mathbb{R}^{n/2 \times k}$, such that $\mathbf{R}_j$ performs pairwise 2D rotations of the input vector using parameters from the $j$th column of $\Theta$, and $\mathbf{Q}_j$ is a fixed permutation matrix. This formulation is inspired by a feed-forward predecessor (MacDonald et al., 2021). It should be noted that, in the publicly available implementation of the VPNN layers from MacDonald et al. (2021) written in Keras (Chollet et al., 2015) that we require to implement SGORNN[1], a modification to the permutation structure of $\mathbf{U}$ is present in the code.[2] This detail is ignored to simplify notation, but is justified in the supplementary material for the sake of proving Corollary 1.

The constraint we propose for the scalar gates of SGORNN to ensure generalization takes the following form:

$$\beta \le 1 - 2\alpha. \tag{3}$$

At first glance, this constraint may seem to limit the representation capacity of the model as FastRNN takes $\beta = 1 - \alpha > 1 - 2\alpha$ by assumption during theoretical analysis. However, typically $\alpha \ll 1$, in which case $\beta + 2\alpha \approx \beta + \alpha$, showing that an approximation to the standard formulation of the scalar gates remains. This constraint is empirically justified in later sections.

Because our proposed constraints on $\alpha$ and $\beta$ are independent of all other parameters (or bounds), they are very simply enforced at training time by applying Equation 3. In practice, the parameter $\beta$ is clipped after applying the sigmoid function, so that it lies in the half-open interval $(0, 1 - 2\alpha]$.

## 2.1 Generalization Bound

Initially, we state the constraint required for the generalization of FastRNN, to contrast with the constraints we require to guarantee the generalization of SGORNN and avoid exploding gradients. It is assumed for FastRNN that $\|\mathbf{U}\|_F \le R_{\mathbf{U}}$ for some $R_{\mathbf{U}} > 0$, and the required constraint takes the form:

$$\alpha \le \min\left(\frac{1}{4T|\mathcal{D}\|\mathbf{U}\|_2 - 1|}, \frac{1}{4TR_{\mathbf{U}}}, \frac{1}{T|\|\mathbf{U}\|_2 - 1|}\right). \tag{4}$$

In the above equation, $\mathcal{D}$ is a real number derived from the parameters and activation function of the model. From this, Kusupati et al. (2019) established an $\mathcal{O}(\alpha T)$ generalization bound for FastRNN which becomes independent of $T$ since $\alpha = \mathcal{O}(1/T)$. However, establishing this constraint in practice is difficult for several reasons. First, the constraint is dependent on the sequence length $T$ of the model, which may be ill-defined in real-world scenarios, and can lead to vanishing $\alpha$ for very long sequences. Second, it relies on restricting the function space to those RNNs with $\|\mathbf{U}\|_F \le R_{\mathbf{U}}$, so that $R_{\mathbf{U}}$ is well-defined for use in Equation 4. In addition, in many cases, choosing such an $R_{\mathbf{U}}$ before starting the training may decrease representation capacity drastically or, again, lead to vanishing $\alpha$. Finally, the constraint becomes undefined for orthogonal matrices ($\|\mathbf{U}\|_2 = 1$). These issues motivate the use of our constraint from Equation 3 for SGORNN, which depends only on the use of the scalar gates $\alpha$ and $\beta$, and can be enforced at training time through a simple clipping operation without any prior assumptions about the sequence length or other parameters of the RNN. We define the class $\mathcal{F}_T$ of SGORNN networks with ReLU activations as follows. Given bounds

---

[1]The implementation of SGORNN and experiment code is provided in a supplementary ZIP archive

[2]See the VPNN Python implementation at https://github.com/wtaylor17/keras-vpnn

$R_{\mathbf{W}}, R_{\mathbf{x}} > 0$, any fixed $\alpha, \beta \in (0, 1)$ with $\beta \leq 1 - 2\alpha$, a mapping $\phi$ from parameters to orthogonal matrices, and an input space $\mathcal{X}$ of sequences $\{\mathbf{x}_t\}_{t=1}^T$ where $\|\mathbf{x}_t\|_2 \leq R_{\mathbf{x}}$, the SGORNN function class is defined as:

$$\mathcal{F}_T = \{(\mathbf{x}_T, \mathbf{h}_{T-1}) \mapsto \alpha\sigma(\mathbf{W}\mathbf{x}_T + \phi(\Theta)\mathbf{h}_{T-1}) + \beta\mathbf{h}_{T-1} : \|\mathbf{W}\|_F \leq R_{\mathbf{W}}, \Theta \in \text{dom}(\phi)\}. \quad (5)$$

It is implicit in the definition of the SGORNN function class that bias vectors are omitted or absorbed into the weights matrix, $\sigma$ is the ReLU activation, and that the dimensions of $\mathbf{h}_{T-1}, \mathbf{x}_T$, and $\mathbf{W}$ are appropriate. The Rademacher complexity of this function class $\mathcal{F}_T$ leads to the following theorem, which describes the generalization ability of SGORNN models with ReLU activation functions and with a single output neuron using any 1-Lipschitz activation function. This provides the equivalent of the FastRNN generalization gap result for our new proposed architecture:

**Theorem 1** *Let $\mathbf{h}_T$ denote the final hidden state of a SGORNN model from the function class $\mathcal{F}_T$ for predetermined bounds $R_{\mathbf{W}}, R_{\mathbf{x}}$. For any vector $\mathbf{v} \in \mathbb{R}^h$ with $\|\mathbf{v}\|_2 \leq R_{\mathbf{v}}$, and any 1-Lipschitz activation function $\sigma_y$, denote $f \in \sigma_y \circ \mathbf{v} \circ \mathcal{F}_T$ as a single-neuron output model such that $f(\mathbf{X}) = \sigma_y(\mathbf{v}^\top \mathbf{h}_T)$. Then, denoting $L : \mathbb{R} \times \mathbb{R} \to [0, B_L]$ as any bounded 1-Lipschitz loss function for some $B_L > 0$, we have with probability at least $1 - \delta$ over the draw of an i.i.d. sample $S = \{(\mathbf{X}^{(i)}, y^{(i)})\}_{i=1}^m$ from any input-output distribution, the following generalization gap holds:*

$$\mathbb{E}[L(y, f(\mathbf{X}))] \leq \frac{1}{m}\sum_{i=1}^m L(y^{(i)}, f(\mathbf{X}^{(i)})) + 4\mathcal{C}\frac{\alpha T}{\sqrt{m}} + 3B_L\sqrt{\frac{\log 2/\delta}{m}} \quad (6)$$

*where $\mathcal{C} = R_{\mathbf{x}}R_{\mathbf{W}}R_{\mathbf{v}}$ represents the boundedness of the model parameters and input data, as in the analysis of FastRNN (Kusupati et al., 2019), and the expectation is taken over the input-output distribution.*

The presented theoretical result for SGORNN shows that the generalization of the model is independent of $\mathbf{U}$ and $\Theta$, which were not used to compute the bounds on $\alpha$ and $\beta$ and do not appear in the definition of $\mathcal{C}$. This is because the spectral norm for matrices is consistent with the Euclidean vector norm and $\|\mathbf{U}\|_2 = 1$ for SGORNN. Further, the constraints on the scalar gates from Equation 3 do not depend on the sequence length $T$ of the model. A proof of Theorem 1 is provided in Appendix A.1.

## 2.2 EXPLODING GRADIENTS

Next, we analyze the gradients $\frac{\partial L}{\partial \mathbf{W}}$ and $\frac{\partial L}{\partial \Theta}$ of a loss function $L$, which is assumed to be 1-Lipschitz. As $\mathbf{W}$ and $\mathbf{U}$ are the only parameter matrices interacting with the loss through temporal layers, we study the maximum entry norm $\|\cdot\|_{max}$ of the loss function with respect to these parameters to show that SGORNN does not suffer from exploding gradients through time when applying a weaker constraint on $\alpha$ and $\beta$ than the one from Equation 3. This analysis leads to our second theorem:

**Theorem 2** *Let $f \in \sigma_y \circ \mathbf{v} \circ \mathcal{F}_T$ be a SGORNN model with ReLU activation functions (as in Theorem 1) with a trivialization mapping $\phi$ satisfying $\left\|\frac{\partial \phi(\Theta)}{\partial \Theta_{ij}}\right\|_{max} \leq M_\phi$ for all indices $i, j$, and with $\beta \leq 1 - \alpha$. Then, the maximum parameter gradients of a 1-Lipschitz loss $L$ satisfy:*

$$\left\|\frac{\partial L}{\partial \Theta}\right\|_{max} \leq \mathcal{C}M_\phi h(\alpha T)^2, \quad (7)$$

$$\left\|\frac{\partial L}{\partial \mathbf{W}}\right\|_{max} \leq \mathcal{C}\alpha T\frac{\sqrt{h}}{R_{\mathbf{W}}}. \quad (8)$$

*Where $\mathcal{C}$ and $R_{\mathbf{W}}$ are defined as in Theorem 1.*

From such a theorem, we obtain the following corollary concerning our chosen implementation of SGORNN:

**Corollary 1** *The VPRNN trivialization satisfies Theorem 2 with $M_\phi = 4\sqrt{h}\log_2(h)$, and so the gradients of the parameters $\Theta$ satisfy:*

$$\left\|\frac{\partial L}{\partial \Theta}\right\|_{max} \leq 4\mathcal{C}h^{3/2}\log_2(h)(\alpha T)^2. \tag{9}$$

The theoretical bounds on the gradients above follow from the gradient computations provided in FastRNN (Kusupati et al., 2019) combined with knowledge of the VPRNN architecture (Taylor-Melanson et al., 2021). The proof of Theorem 2 is provided in Appendix A.2, while the details of Corollary 1 are in Appendix A.3.

It should be noted that the above bounds differ from the findings of Arjovsky et al. (2016), who provided gradient bounds on general unitary RNN models independent of $T$. This is because the gradient computation of FastRNN (and consequently SGORNN) is greatly influenced by the residual connection through $\alpha$ and $\beta$. Regardless of this complication, we have shown that SGORNN does not suffer from the exponentially exploding gradients often seen when training RNNs.

## 3 EXPERIMENTS

### 3.1 ADDITION PROBLEM

As a proof of concept, the first experiment considered was the addition problem for RNNs, which was first proposed by Hochreiter & Schmidhuber (1997). This problem takes the form of a regression task, typically on the output vector $\mathbf{h}_T$ of a recurrent network. The input sequence $\mathbf{x}_t$ is a 2D sequence $\mathbf{x}_t = \begin{pmatrix} u_t & c_t \end{pmatrix}^\top$, with $u_t$ being uniformly distributed in $[0,1]$ (i.i.d.) and $c_t$ representing a binary sequence. This binary sequence is mostly filled with 0s, having only two elements at positions $i$ and $j$ set as 1, in which $i$ is chosen uniformly in $[1, T/2]$ and $j$ chosen uniformly in $(T/2, T]$. Intuitively, the addition problem gives the sum of two random target positions in the sequence, that the RNN model must recognize and maintain (remember) over time.

The addition problem has been used as a benchmark for many RNN models which aim to process long sequences, as the randomly generated data introduces long-term dependencies (Li et al., 2018; Taylor-Melanson et al., 2021; Hochreiter & Schmidhuber, 1997; Arjovsky et al., 2016). A model applied on the addition problem is considered converged if it can consistently and considerably beat the baseline (MSE) error of approximately $0.167$ on unseen data (Taylor-Melanson et al., 2021; Li et al., 2018), which is the error when the mean of the target ($y = 1$) is always predicted. In our experiments, all data is randomly generated as performed in the recent literature (Taylor-Melanson et al., 2021; Li et al., 2018). The regression target $y$ over the sequence is the sum of the two random numbers as indicated by the indices $i, j$, defined mathematically as:

$$y(\mathbf{X}) = \sum_{t=1}^{T} u_t c_t = u_i + u_j. \tag{10}$$

We compared the proposed SGORNN with VPRNN, FastRNN, and LSTM using the same model configuration and batch size for all architectures. In this experiment, all models had a hidden state dimension of $128$, and the SGORNN and VPRNN used $k = 14$ rotational sublayers, which is the maximum number allowed based on the constraint from VPRNN architecture (Taylor-Melanson et al., 2021). Every model consisted of just one RNN layer with ReLU as the activation function. The constraint from Equation 3 was imposed on the SGORNN model so that the model satisfies Theorem 1. The RMSprop optimizer (Tieleman & Hinton, 2012) was used, in which the learning rate was decayed linearly to zero during training. Similarly to experiments conducted by Taylor-Melanson et al. (2021), models were trained with a batch size of $64$ for a maximum of $20,000$ training steps, and 10 batches of unseen data were used to record validation MSE after every 100 training steps. We note that all models trained by Li et al. (2018) that converged on the addition problem did so within $20,000$ training steps, motivating our choice of cutoff for measuring convergence.

The goal of this experiment was to show which of the models tested are able to converge on different sequence lengths. For this reason, initial learning rates of each model were first set to $10^{-2}$, and were

decreased by a factor of 10 each time a model could not converge. For example, if a model could not converge with the initial learning rate of $10^{-2}$ it was retrained with an initial learning rate of $10^{-3}$. This was allowed to be done twice for each model, so that all initial learning rates came from the set $\{10^{-2}, 10^{-3}, 10^{-4}\}$. In this way, we performed a basic search for the maximum learning rate to allow the convergence of the models. However, due to issues such as exploding gradients, some models could not converge for this task. In the case of partial convergence (e.g. VPRNN with $T = 1000$, see Figure 2), a learning rate decrease was performed to ensure better convergence could not be obtained. For such a scenario, the result for the best learning rate was reported. Figures 1, 2, and 3 present the validation error obtained for each model using the sequence lengths 500, 1000, and 5000 respectively.

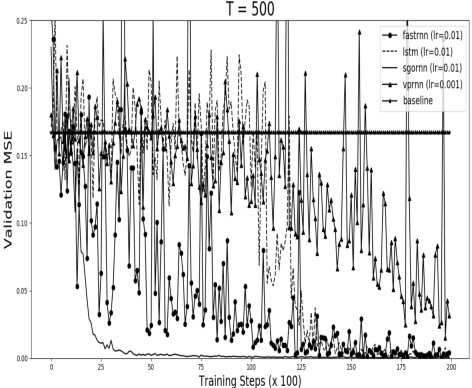

Figure 1: Addition problem results for the three models tested with $T = 500$.

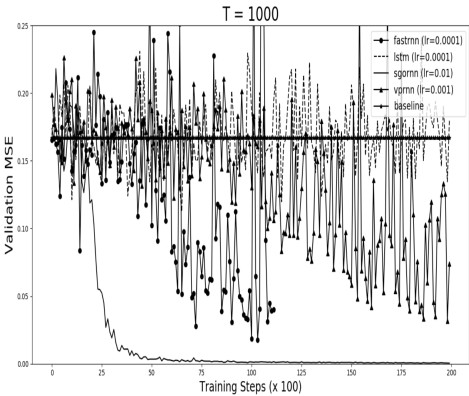

Figure 2: Addition problem results for the three models tested with $T = 1000$.

For the shortest sequence length, $T = 500$, all three models were able to converge using the largest initial learning rate of $10^{-2}$, though FastRNN and SGORNN provided the most efficient convergence. In the case of sequences of length $T = 1000$, VPRNN failed to converge with learning rates of $10^{-2}$ and $10^{-4}$, though it achieved partial convergence with a learning rate of $10^{-3}$. Also with $T = 1000$, SGORNN converged efficiently with the largest learning rate of $10^{-2}$. FastRNN suffered from exploding gradients and divergent loss using all learning rates, even with the smallest learning rate of $10^{-4}$. FastRNN also achieved partial convergence for $T = 1000$ as shown in Figure 2, but was unable to reach an efficient solution before exploding gradients occurred. Lastly, for the largest sequence length of $T = 5000$, we observed that FastRNN suffered from exploding gradients and consequently a divergent MSE loss for all initial learning rates. VPRNN also failed to converge for $T = 5000$ regardless of the initial learning rate chosen, though it did not suffer from exploding gradients. In contrast to this result, SGORNN still converged efficiently using the largest initial learning rate of $10^{-2}$ and did not suffer from exploding gradients. The LSTM model failed for all sequence lengths greater than $T = 500$ regardless of the chosen learning rate.

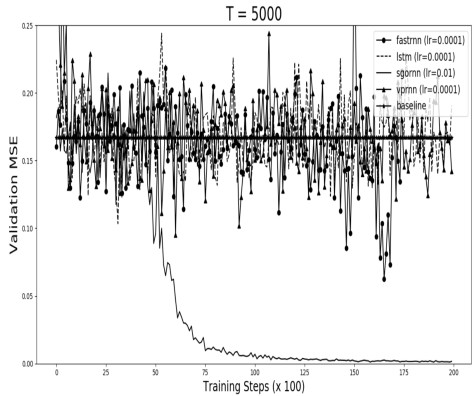

Figure 3: Addition problem results for the two previously convergent models, with $T = 5000$.

Table 1: Parameter counts and performance of the three RNN models tested on the addition problem. Partial convergence for VPRNN and FastRNN is indicated by parentheses.

| Model Name | Parameter Count | Max Sequence Length | Exploding Gradients |
|---|---|---|---|
| VPRNN | **1,409** | 1000 (partial) | Never |
| SGORNN | 1,411 | **5,000** | Never |
| FastRNN | 16,899 | 1,000 (partial) | $T \geq 1000$ |
| LSTM | 67,201 | 500 | Never |

## 3.2 HAR-2 Classification

The HAR-2 classification task is a binary classification task derived from a human movement data set collected using smartphones (Anguita et al., 2013). The data set consists of 9-dimensional sequences of length $T = 128$, having $7,352$ training instances and $2,947$ testing instances. We used this dataset to train and evaluate SGORNN, FastRNN, VPRNN, and LSTM models, conducting a performance comparison between them. All models were single-layer RNNs with $128$ hidden units, except for the LSTM which was chosen to have $64$ hidden units to roughly match the number of parameters of the FastRNN, similar to how hidden state sizes were chosen for a language modelling task in GORU (Jing et al., 2019). The SGORNN and VPRNN models had $k = 14$ rotational sublayers, which is the maximum allowed following from the constraint presented by Taylor-Melanson et al. (2021). We used an RMSprop optimizer (Tieleman & Hinton, 2012) with an initial learning rate of $10^{-3}$. Models were trained for 300 epochs using a batch size of $100$, and the learning rate was decreased by a factor of $10$ after $200$ epochs. This strategy is similar to the one employed for FastRNN (Kusupati et al., 2019) with the main difference being our lack of hyperparameter tuning. We trained each model 10 times using different random seeds.[3] The constraint from Equation 3 was imposed on the SGORNN model to ensure good generalization based on Theorem 1. This experiment shows that our proposed constraint from Equation 3 can be applied in practice using a real-world data set.

As shown in Table 2, SGORNN achieved the highest mean test accuracy among the three models trained. Although this accuracy is not drastically above that of FastRNN or LSTM, across the 10 training runs, the standard deviation of the test accuracies measured was the lowest for SGORNN. This suggests that SGORNN had both greater performance and more stable training than the other models, while requiring far fewer parameters than FastRNN or LSTM.

We use the difference between the training and test performance (for both accuracy and loss) to estimate the degree of overfitting of the models trained on this task. Plots of these estimates of overfitting are illustrated in Figure 4. Interestingly, VPRNN has the largest gap between training and testing accuracy, although this gap does not significantly differ from that of FastRNN or LSTM.

---

[3]To simplify reproducibility, the seed for run $i$ was made to be the integer $i \in \{1, 2, \ldots, 10\}$.

Table 2: Parameter counts and accuracy of the three RNN models tested on the HAR-2 classification problem, including the standard deviation across 10 training runs.

| Model Name | Parameter Count | Test Accuracy ($\pm$ Standard Deviation) |
|---|---|---|
| VPRNN | **2,305** | $92.29 \pm 0.96\%$ |
| SGORNN | 2,307 | **$94.07 \pm 0.2$**$\%$ |
| FastRNN | 17,795 | $93.77 \pm 0.49\%$ |
| LSTM | 19,009 | $93.45 \pm 1.22\%$ |

Further, the accuracy gap for both FastRNN, VPRNN, and LSTM seems to increase as training goes on, while the gap for SGORNN seems to decrease and eventually converge to a stationary value between 3% and 4% (Figure 4a). Even more interesting is the degree of overfitting of FastRNN with respect to the entropy loss, where the decrease in learning rate after 200 epochs caused the gap to grow much faster. VPRNN, FastRNN, and LSTM have an entropy gap growing with the number of epochs, while the SGORNN model loss gap remains nearly constant and has a very small variance (Figure 4b). This behavior suggests that, as training goes on, the degree of overfitting of SGORNN decreases or remains constant while it increases for all other models.

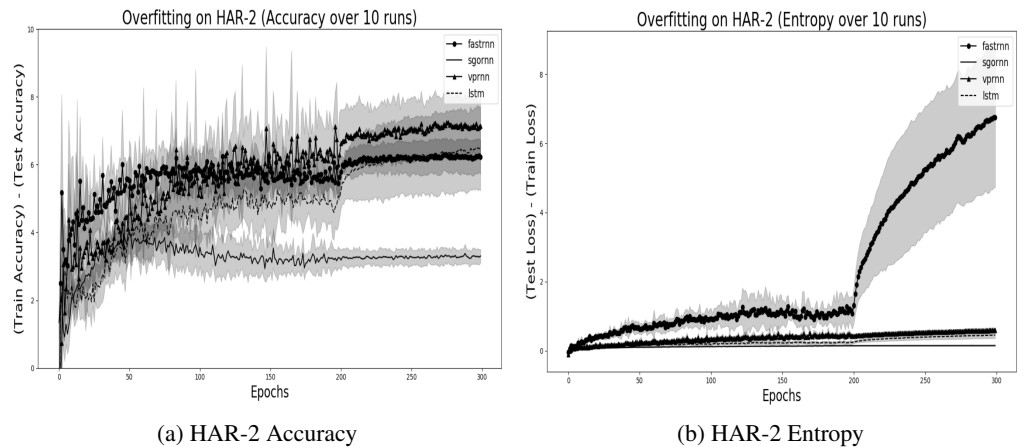

(a) HAR-2 Accuracy          (b) HAR-2 Entropy

Figure 4: The observed estimate of overfitting (difference between train/test performance) for the three models trained on HAR-2. Solid lines indicate the 10-run mean, and the standard deviation over 10 runs is displayed using shaded regions.

## 3.3 PENN TREEBANK LANGUAGE MODELLING

The Penn Treebank (PTB) data set is an annotated corpus for natural language processing tasks (Marcus et al., 1993). The word-level version of the data set consists of approximately $929,000$ training tokens, $74,000$ validation tokens, and $83,000$ testing tokens. The most common $10,000$ tokens are used, with the rest of the tokens replaced by <unk> tokens. Newline characters are replaced with <eos> tokens. We use this data set to evaluate our models on a language modelling task, where the output of the RNN at time $t$ is a probability distribution over the token to be observed at time $t + 1$ calculated using a softmax activation function on a fully-connected output layer.

We take our sequence length and model architecture from FastRNN (Kusupati et al., 2019), processing sequences of length $300$ and fixing the dimension of the trainable embeddings of each model to be the same as that of the hidden states of the RNN being trained. Our models were all single-cell RNNs with 256 hidden units. Because the classification layer of these language models accounts for such a large proportion of the overall parameters, the LSTM model was allowed to remain at 256 hidden units (unlike in the previous HAR-2 task). The SGORNN and VPRNN models had $k = 16$ rotational sublayers. The constraint from Equation 3 was enforced on SGORNN to ensure generalization according to Theorem 1. We used the RMSprop optimizer (Tieleman & Hinton, 2012), with the same number of epochs and learning rate decay scheme as with the HAR-2 experiment

(Section 3.2). As an attempt to prevent overfitting in all models, we applied dropout of $0.3$ to the embeddings given as input to the RNN cells. The metric used to evaluate models is the test perplexity, which measures the level of uncertainty of the language models (lower is better). It is seen in Table 3 that SGORNN outperforms both of its predecessor models, and shows comparable performance to LSTM despite having only a small fraction of the number of parameters within its RNN cell.

Table 3: Parameter counts and test perplexity of the RNN models trained on the PTB data set for word-level language modelling. Parameter counts only include the size of the RNN cells.

| Model Name | Parameter Count | Test Perplexity |
|---|---|---|
| VPRNN | **67,840** | 257.88 |
| SGORNN | 67,842 | 132.26 |
| FastRNN | 131,330 | 146.75 |
| LSTM | 525,312 | **126.78** |

## 4 CONCLUSIONS

This paper has presented a new RNN architecture which combines orthogonal constraints with a scalar gating mechanism. This new RNN, which we call SGORNN, was implemented for the purpose of experimentation using an orthogonal transformation scheme from recent literature called VPRNN. However, as only Corollary 1 relies directly on the structure of the VPRNN mapping, our main theoretical results from Theorems 1 and 2 can be applied with different mapping strategies, such as the EXPRNN mapping (Casado & Martínez-Rubio, 2019).

Our theoretical analysis presented both bounds on the gradients of SGORNN and a generalization gap bound for Lipschitz loss functions. The gradient bound for SGORNN shows that gradients do not explode exponentially through time, but instead are $\mathcal{O}((\alpha T)^2)$ in the worst case when $\alpha \leq 1-\beta$. This constraint to control gradients is weaker than the one required for generalization of our architecture, in that enforcing the generalization constraint also enforces the gradient bound constraint. In theory, through further controlling bounds on $\alpha$, we can fully control the upper gradient bound presented. The constraint on $\alpha$ (Equation 3) depends only on the other scalar gate $\beta$, rather than the $\alpha = \mathcal{O}(1/T)$ constraint required for generalization of a FastRNN, which depends on bounds constructed for other trainable parameters. Because of this dependence, our constraint for generalization was easily enforced through a clipping operation for the three tasks in our experiments.

In our experiments, we first trained models on the addition problem, where SGORNN was able to converge on sequences of length $T = 5000$ using less than $1,500$ parameters, while FastRNN suffered from exploding gradients and both VPRNN and LSTM failed to converge outright. This is the smallest number of parameters that was found in the literature for RNNs able to process such long sequences on the addition problem. Further, it shows that SGORNN may require just one RNN cell to perform accurately on tasks where other architectures with a reduced number of parameters such as IndRNN (Li et al., 2018) and VPRNN (Taylor-Melanson et al., 2021) have been noted to require two stacked cells. On the HAR-2 classification task, the SGORNN model had both the highest accuracy and suffered from the least amount of overfitting of all models considered. On the PTB word-level language modelling task, SGORNN outperformed both VPRNN and FastRNN in terms of test perplexity, and shows comparable performance to our LSTM. It is worth noting that our test perplexity for FastRNN is higher than that reported by Kusupati et al. (2019), which may be attributed to differences in hyperparameters. Based on the experiments, we conclude that the SGORNN architecture brings enforceable generalization and well-controlled gradients to the FastRNN architecture, and brings increased representation capacity to orthogonal RNN architectures such as VPRNN, reinforcing our theoretical results.

Future work on SGORNN will include further investigation of generalization ability, as well as investigation of alternative constraints on the scalar gates. Additional architectures such as SGORNN models with standard gating mechanisms may also be investigated. Finally, convergence analysis of SGORNN and the investigation of mappings to orthogonal matrices other than that of VPRNN will also be the topic of future work.

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

## A    PROOFS OF STATED THEOREMS AND DETAILS OF COROLLARIES

In the following proofs, we follow notation similar to (Kusupati et al., 2019). This is different from that of (Taylor-Melanson et al., 2021) who denote the recurrent weights matrix by $\mathbf{V}$ rather than $\mathbf{U}$. We choose to denote the empirical Rademacher complexity of a function class $\mathcal{F}$ by $\mathcal{R}_m(\mathcal{F})$, to indicate that the average is taken over a random sample of size $m$. This differs slightly from both (Kusupati et al., 2019) and (Bousquet et al., 2003) who use $n$ instead of $m$ to denote the size of the random sample, since we use $n$ to describe the input data dimension in Section 2.

### A.1    PROOF OF THEOREM 1

We first begin with a description of the one-sided Rademacher complexity bound framework used. For any probability distribution $D$ over input-output pairs $z = (\mathbf{X}, y) \in \mathcal{Z}$, any bound $B_L > 0$, and any loss function $L : \mathcal{Z} \times \hat{\mathcal{Y}} \to [0, B_L]$ taking in a training instance and a prediction $\hat{y} \in \hat{\mathcal{Y}}$, define a class of functions over any class of predictors $\mathcal{F}$ taking values in $\hat{\mathcal{Y}}$ as:

$$L \circ \mathcal{F} = \{\ell_f = (\mathbf{X}, y) \mapsto L(y, f(\mathbf{X})) : f \in \mathcal{F}\}, \tag{11}$$

so that if $\ell_f \in L \circ \mathcal{F}$ then $\ell_f : \mathcal{Z} \to [0, B_L]$ is a mapping from input-output pairs to loss values. The empirical Rademacher complexity of this class of functions is defined as in (Bousquet et al., 2003) on a sample $S = \{z^{(1)}, \ldots, z^{(m)}\}$ drawn i.i.d. from $D$ as:

$$\mathcal{R}_m(L \circ \mathcal{F}) = \mathbb{E}_\epsilon \left[ \sup_{\ell_f \in L \circ \mathcal{F}} \frac{1}{m} \sum_{i=1}^m \epsilon_i \ell_f(z^{(i)}) \right] \tag{12}$$

The main result we require to prove the generalization of SGORNN is the one-sided Rademacher complexity bound from Bartlett & Mendelson (2002), which is also used by FastRNN (Kusupati et al., 2019). It states that, for all $\delta > 0$, with probability at least $1 - \delta$ over the draw of $S$, for any $\ell_f \in L \circ \mathcal{F}$ and distribution $D$ over $\mathcal{Z}$:

$$\mathbb{E}_{z \sim D}[\ell_f(z)] \leq \frac{1}{m} \sum_{i=1}^m \ell_f(z^{(i)}) + 2\mathcal{R}_m(L \circ \mathcal{F}) + 3B_L \sqrt{\frac{\log 2/\delta}{2m}} \tag{13}$$

**Theorem 1** *Let $\mathbf{h}_T$ denote the final hidden state of a SGORNN model from the function class $\mathcal{F}_T$ for predetermined bounds $R_{\mathbf{W}}, R_{\mathbf{x}}$. For any vector $\mathbf{v} \in \mathbb{R}^h$ with $\|\mathbf{v}\|_2 \leq R_{\mathbf{v}}$, and any 1-Lipschitz activation function $\sigma_y$, denote $f \in \sigma_y \circ \mathbf{v} \circ \mathcal{F}_T$ as a single-neuron output model such that $f(\mathbf{X}) = \sigma_y(\mathbf{v}^\top \mathbf{h}_T)$. Then, denoting $L : \mathbb{R} \times \mathbb{R} \to [0, B_L]$ as any bounded 1-Lipschitz loss function for some $B_L > 0$, we have with probability at least $1 - \delta$ over the draw of an i.i.d. sample $S = \{(\mathbf{X}^{(i)}, y^{(i)})\}_{i=1}^m$ from any input-output distribution, the following generalization gap holds:*

$$\mathbb{E}[L(y, f(\mathbf{X}))] \leq \frac{1}{m} \sum_{i=1}^m L(y^{(i)}, f(\mathbf{X}^{(i)})) + 4\mathcal{C} \frac{\alpha T}{\sqrt{m}} + 3B_L \sqrt{\frac{\log 2/\delta}{m}} \tag{14}$$

*where $\mathcal{C} = R_{\mathbf{x}} R_{\mathbf{W}} R_{\mathbf{v}}$ represents the boundedness of the model parameters and input data, as in the analysis of FastRNN (Kusupati et al., 2019), and the expectation is taken over the input-output distribution.*

It just remains to establish a bound on $\mathcal{R}_m(L \circ \mathcal{F})$. As before, denote $f(\mathbf{X}^{(i)}) = \sigma_y(\mathbf{v}^\top \mathbf{h}_T^{(i)})$ as the output of the model on a given input sequence $\mathbf{X}^{(i)} = (\mathbf{x}_1^{(i)}, \ldots, \mathbf{x}_T^{(i)})$, where $\mathbf{h}_T^{(i)}$ denotes the final hidden state of the SGORNN on the given input. Denote the class of SGORNN networks (outputting $\mathbf{h}_T$) as $\mathcal{F}_T$, so that $\mathcal{F} = \sigma_y \circ \mathbf{v} \circ \mathcal{F}_T$ is the overall class of hypothesis functions. Because both $L$ and $\sigma_y$ are assumed to be 1-Lipschitz, we have by the Ledoux-Talagrand contraction (the needed version of which has a proof in (Mohri & Medina, 2014)) that:

$$\mathcal{R}_m(L \circ \mathcal{F}) \leq \mathcal{R}_m(\mathcal{F}) = \mathcal{R}_m(\sigma_y \circ \mathbf{v} \circ \mathcal{F}_T) \leq \mathcal{R}_m(\mathbf{v} \circ \mathcal{F}_T). \tag{15}$$

By definition of the empirical Rademacher complexity, we then have that:

$$\mathcal{R}_m(\mathbf{v} \circ \mathcal{F}_T) = \mathbb{E}_\epsilon \left[ \sup_{\mathbf{v},\mathbf{W},\Theta} \frac{1}{m} \sum_{i=1}^m \epsilon_i \mathbf{v}^\top \mathbf{h}_T^{(i)} \right]. \tag{16}$$

Next, recall that $\|\mathbf{v}\|_2 \leq R_{\mathbf{v}}$ and $\|\mathbf{W}\|_F \leq R_{\mathbf{W}}$. There is no explicit restriction placed on $\Theta$ as in the definition of $\mathcal{F}_T$. Note that:

$$\mathcal{R}_m(\mathbf{v} \circ \mathcal{F}_T) = \mathbb{E}_\epsilon \left[ \sup_{\mathbf{v},\mathbf{W},\Theta} \frac{1}{m} \mathbf{v}^\top \sum_{i=1}^m \epsilon_i \mathbf{h}_T^{(i)} \right] \tag{17}$$

$$\leq \mathbb{E}_\epsilon \left[ \sup_{\mathbf{v},\mathbf{W},\Theta} \frac{1}{m} \|\mathbf{v}\|_2 \left\| \sum_{i=1}^m \epsilon_i \mathbf{h}_T^{(i)} \right\|_2 \right] \tag{18}$$

$$\leq R_{\mathbf{v}} \mathbb{E}_\epsilon \left[ \sup_{\mathbf{W},\Theta} \frac{1}{m} \left\| \sum_{i=1}^m \epsilon_i \mathbf{h}_T^{(i)} \right\|_2 \right]. \tag{19}$$

Now, we follow a similar strategy employed for FastRNN (Kusupati et al., 2019). Denoting $\mathcal{R}_m(\mathcal{F}_T)$ as the above expected value, we have modified the notation of an inequality for $\mathcal{R}_m(\mathcal{F}_T)$, provided by Kusupati et al. (2019), to capture the supremum over recurrent parameters $\Theta$:

$$\mathcal{R}_m(\mathcal{F}_T) \leq \beta \mathcal{R}_m(\mathcal{F}_{T-1}) + \frac{2\alpha}{m} \left( \mathbb{E}_\epsilon \left[ \sup_{\mathbf{W}} \left\| \sum_{i=1}^m \epsilon_i \mathbf{W} \mathbf{x}_T^{(i)} \right\|_2 \right] + \mathbb{E}_\epsilon \left[ \sup_{\mathbf{W},\Theta} \left\| \sum_{i=1}^m \epsilon_i \mathbf{U} \mathbf{h}_{T-1}^{(i)} \right\|_2 \right] \right). \tag{20}$$

Equation 20 relies on an analog of the Ledoux-Talagrand contraction to remove dependence on the activation function $\sigma$. The first expected value in this equation is bounded by $R_{\mathbf{W}} R_{\mathbf{x}} \sqrt{m}$ using the same argument as Kusupati et al. (2019). Because $\mathbf{U}$ is orthogonal, the second expected value in Equation 20 is:

$$\mathbb{E}_\epsilon \left[ \sup_{\mathbf{W},\Theta} \left\| \sum_{i=1}^m \epsilon_i \mathbf{U} \mathbf{h}_{T-1}^{(i)} \right\|_2 \right] = \mathbb{E}_\epsilon \left[ \sup_{\mathbf{W},\Theta} \left\| \mathbf{U} \sum_{i=1}^m \epsilon_i \mathbf{h}_{T-1}^{(i)} \right\|_2 \right]$$

$$= \mathbb{E}_\epsilon \left[ \sup_{\mathbf{W},\Theta} \left\| \sum_{i=1}^m \epsilon_i \mathbf{h}_{T-1}^{(i)} \right\|_2 \right]$$

$$= m \mathcal{R}_m(\mathcal{F}_{T-1})$$

So that, by substituting the bounds on the expected values, Equation 20 becomes:

$$\mathcal{R}_m(\mathcal{F}_T) \leq (\beta + 2\alpha) \mathcal{R}_m(\mathcal{F}_{T-1}) + \frac{2\alpha R_{\mathbf{W}} R_{\mathbf{X}}}{\sqrt{m}} \tag{21}$$

Now, using the fact that $\beta \leq 1 - 2\alpha$, the coefficient of $\mathcal{R}_m(\mathcal{F}_{T-1})$ in the above recurrence is at most 1. Assuming $\mathbf{h}_0^{(i)} = \mathbf{0}$ for all $i$, we get that $\mathcal{R}_m(\mathcal{F}_0) = 0$ and consequently:

$$\mathcal{R}_m(\mathcal{F}_T) \leq 2\alpha T \frac{R_\mathbf{W} R_\mathbf{x}}{\sqrt{m}}. \tag{22}$$

Setting $\mathcal{C} = R_\mathbf{v} R_\mathbf{W} R_\mathbf{x}$, combining Equation 19 and Equation 22 gives a bound on the Rademacher complexity of $L \circ \mathcal{F}$ of:

$$\mathcal{R}_m(L \circ \mathcal{F}) \leq 2\mathcal{C} \frac{\alpha T}{\sqrt{m}}, \tag{23}$$

and finally, the desired bound holding with probability at least $1 - \delta$ comes from combining Equations 13 and 23:

$$\mathbb{E}_{z \sim D}[\ell_f(z)] \leq \frac{1}{m} \sum_{i=1}^m \ell_f(z^{(i)}) + 4\mathcal{C} \frac{\alpha T}{\sqrt{m}} + 3B_L \sqrt{\frac{\log 2/\delta}{2m}}, \tag{24}$$

which is equivalent to the original statement of the theorem, as $\ell_f(z)$ is simply shorthand for $L(y, f(\mathbf{X}))$. $\blacksquare$

### A.2 PROOF OF THEOREM 2

The proof relies on the two following formulas, taken from the formulation presented by Kusupati et al. (2019):

$$\frac{\partial L}{\partial \mathbf{U}} = \alpha \sum_{t=1}^T \mathbf{D}_t \left( \prod_{k=t}^{T-1} (\alpha \mathbf{U}^\top \mathbf{D}_{k+1} + \beta \mathbf{I}) \right) (\nabla_{\mathbf{h}_T} L) \mathbf{h}_{t-1}^\top, \tag{25}$$

$$\frac{\partial L}{\partial \mathbf{W}} = \alpha \sum_{t=1}^T \mathbf{D}_t \left( \prod_{k=t}^{T-1} (\alpha \mathbf{U}^\top \mathbf{D}_{k+1} + \beta \mathbf{I}) \right) (\nabla_{\mathbf{h}_T} L) \mathbf{x}_t^\top, \tag{26}$$

where $\mathbf{D}_k = \text{diag}(\sigma'(\mathbf{W}\mathbf{x}_k + \mathbf{U}\mathbf{h}_{k-1}))$ is the Jacobian matrix of the pointwise activation function $\sigma$. Note that for ReLU, $\mathbf{D}_k$ is a diagonal matrix with a binary vector on the diagonal.

**Theorem 2** *Let $f \in \sigma_y \circ \mathbf{v} \circ \mathcal{F}_T$ be a SGORNN model with ReLU activation functions (as in Theorem 1) with a trivialization mapping $\phi$ satisfying $\left\| \frac{\partial \phi(\Theta)}{\partial \Theta_{ij}} \right\|_{max} \leq M_\phi$ for all indices $i, j$, and with $\beta \leq 1 - \alpha$. Then, the maximum parameter gradients of a 1-Lipschitz loss $L$ satisfy:*

$$\left\| \frac{\partial L}{\partial \Theta} \right\|_{max} \leq \mathcal{C} M_\phi h(\alpha T)^2, \tag{27}$$

$$\left\| \frac{\partial L}{\partial \mathbf{W}} \right\|_{max} \leq \mathcal{C} \alpha T \frac{\sqrt{h}}{R_\mathbf{W}}. \tag{28}$$

*Where $\mathcal{C}$ and $R_\mathbf{W}$ are defined as in Theorem 1.*

We begin by establishing the first bound involving $\Theta$. Note that by the chain rule:

$$\frac{\partial L}{\partial \Theta_{ij}} = \sum_{\tau, \mu} \frac{\partial L}{\partial \mathbf{U}_{\tau\mu}} \frac{\partial \mathbf{U}_{\tau\mu}}{\partial \Theta_{ij}}. \tag{29}$$

Note also that we simply denoted $\mathbf{U} = \phi(\Theta)$, so:

$$\frac{\partial L}{\partial \Theta_{ij}} = \sum_{\tau, \mu} \frac{\partial L}{\partial \mathbf{U}_{\tau\mu}} \frac{\partial \phi(\Theta)_{\tau\mu}}{\partial \Theta_{ij}}. \tag{30}$$

Thus, since $\left|\frac{\partial \phi(\Theta)_{\tau\mu}}{\partial \Theta_{ij}}\right| \leq M_\phi$:

$$\left|\frac{\partial L}{\partial \Theta_{ij}}\right| \leq M_\phi \sum_{\tau,\mu} \left|\frac{\partial L}{\partial \mathbf{U}_{\tau\mu}}\right| = M_\phi \left\|\frac{\partial L}{\partial \mathbf{U}}\right\|_{1,1} \leq M_\phi h \left\|\frac{\partial L}{\partial \mathbf{U}}\right\|_2, \tag{31}$$

where we use the fact that $\|\mathbf{X}\|_{1,1} \leq \sqrt{h} \|\mathbf{X}\|_F$ and $\|\mathbf{X}\|_F \leq \sqrt{h} \|\mathbf{X}\|_2$ for any $h \times h$ matrix $\mathbf{X}$. Next, we use the subadditivity and submultiplicativity of the spectral norm applied to Eq. 25:

$$\left\|\frac{\partial L}{\partial \mathbf{U}}\right\|_2 \leq \alpha \sum_{t=1}^T \|\mathbf{D}_t\|_2 \left(\prod_{k=t}^{T-1} \left(\alpha \left\|\mathbf{U}^\top\right\|_2 \|\mathbf{D}_{k+1}\|_2 + \beta \|\mathbf{I}\|_2\right)\right) \left\|(\nabla_{\mathbf{h}_T} L)\mathbf{h}_{t-1}^\top\right\|_2.$$

Because ReLU is 1-Lipschitz, $\|\mathbf{D}_k\|_2 \leq 1$ for any $k$. Further, $\left\|\mathbf{U}^\top\right\|_2 = \|\mathbf{I}\|_2 = 1$. Thus, the product in the formula has an upper bound of $(\alpha+\beta)^{T-t} \leq 1$. We also have that $\left\|(\nabla_{\mathbf{h}_T} L)\mathbf{h}_{t-1}^\top\right\|_2 \leq \|\nabla_{\mathbf{h}_T} L\|_2 \|\mathbf{h}_{t-1}\|_2$. Thus:

$$\left\|\frac{\partial L}{\partial \mathbf{U}}\right\|_2 \leq \alpha \|\nabla_{\mathbf{h}_T} L\|_2 \sum_{t=1}^T \|\mathbf{h}_{t-1}\|_2.$$

It is then left to bound $\|\nabla_{\mathbf{h}_T} L\|_2$ and $\|\mathbf{h}_{t-1}\|_2$. First, for the hidden state:

$$\begin{aligned}
\|\mathbf{h}_{t-1}\|_2 &\leq \|\alpha\sigma(\mathbf{W}\mathbf{x}_{t-1} + \mathbf{U}\mathbf{h}_{t-2}) + \beta\mathbf{h}_{t-2}\|_2 \\
&\leq \alpha \|\sigma(\mathbf{W}\mathbf{x}_{t-1} + \mathbf{U}\mathbf{h}_{t-2})\|_2 + \beta \|\mathbf{h}_{t-2}\|_2 \\
&\leq \alpha \|\mathbf{W}\mathbf{x}_{t-1} + \mathbf{U}\mathbf{h}_{t-2}\|_2 + \beta \|\mathbf{h}_{t-2}\|_2 \\
&\leq \alpha(R_\mathbf{W} R_\mathbf{x} + \|\mathbf{U}\mathbf{h}_{t-2}\|_2) + \beta \|\mathbf{h}_{t-2}\|_2 \\
&\leq \alpha R_\mathbf{W} R_\mathbf{x} + (\alpha + \beta) \|\mathbf{h}_{t-2}\|_2 \\
&\leq \alpha R_\mathbf{W} R_\mathbf{x} + \|\mathbf{h}_{t-2}\|_2.
\end{aligned}$$

Solving the above recurrence gives $\|\mathbf{h}_{t-1}\|_2 \leq \alpha T R_\mathbf{W} R_\mathbf{x}$. Next for the gradient, assuming a 1-Lipschitz loss, we get that:

$$\left|\frac{\partial L}{\partial (\mathbf{h}_T)_j}\right| = \left|\frac{\partial L}{\partial \hat{y}} \frac{\partial \hat{y}}{\partial (\mathbf{h}_T)_j}\right| = \left|\frac{\partial L}{\partial \hat{y}} \sigma_y'(\mathbf{v}^\top\mathbf{h}_T) \frac{\partial (\mathbf{v}^\top\mathbf{h}_T)}{\partial (\mathbf{h}_T)_j}\right| \leq |\mathbf{v}_j|. \tag{32}$$

So that $\|\nabla_{\mathbf{h}_T} L\|_2 \leq R_\mathbf{v}$. Plugging these back in gives:

$$\left\|\frac{\partial L}{\partial \mathbf{U}}\right\|_2 \leq R_\mathbf{W} R_\mathbf{x} R_\mathbf{v} (\alpha T)^2 = \mathcal{C}(\alpha T)^2,$$

and consequently the gradient bound:

$$\left|\frac{\partial L}{\partial \Theta_{ij}}\right| \leq \mathcal{C} M_\phi h (\alpha T)^2 \tag{33}$$

as desired (the statement regarding the max norm follows since we chose arbitrary $i, j$). Moving on to the bound on the gradient for $\mathbf{W}$, we get using the same strategy that:

$$\left\|\frac{\partial L}{\partial \mathbf{W}}\right\|_{\max} \leq \left\|\frac{\partial L}{\partial \mathbf{W}}\right\|_{F} \leq \sqrt{h}\left\|\frac{\partial L}{\partial \mathbf{W}}\right\|_{2}$$

$$\leq \sqrt{h}\alpha \left\|\nabla_{\mathbf{h}_T} L\right\|_2 \sum_{t=1}^{T} \|\mathbf{x}_t\|_2$$

$$\leq \alpha T R_{\mathbf{v}} R_{\mathbf{x}} \sqrt{h} = \mathcal{C}\alpha T \frac{\sqrt{h}}{R_{\mathbf{W}}}$$

as desired. ∎

### A.3 DETAILS OF COROLLARY 1

Taylor-Melanson et al. (2021) showed that using the VPRNN trivialization, for any $\Theta$ and $\Delta\Theta$:

$$\|\phi(\Theta) - \phi(\Theta + \Delta\Theta)\|_2 \leq k \|\Delta\Theta\|_2, \tag{34}$$

where $k = 2\lceil \log_2 h \rceil$. Note that $k \leq 4 \log_2 h$ for $h \geq 2$ which can be safely assumed. Now, consider a perturbation $\Delta\Theta$ with zero entries everywhere but the $i,j$th entry, so that:

$$\left|\frac{\partial\phi(\Theta)_{\tau\mu}}{\partial\Theta_{ij}}\right| = \lim_{\Delta\Theta_{ij}\to 0} \frac{|\phi(\Theta)_{\tau\mu} - \phi(\Theta + \Delta\Theta)_{\tau\mu}|}{|\Delta\Theta_{ij}|}. \tag{35}$$

The numerator has the upper bound:

$$|\phi(\Theta)_{\tau\mu} - \phi(\Theta + \Delta\Theta)_{\tau\mu}| \leq \|\phi(\Theta) - \phi(\Theta + \Delta\Theta)\|_F \leq \sqrt{h}\|\phi(\Theta) - \phi(\Theta + \Delta\Theta)\|_2,$$

while the denominator has the lower bound (since $\Delta\Theta$ has only one nonzero entry):

$$|\Delta\Theta_{ij}| = \|\Delta\Theta\|_F \geq \|\Delta\Theta\|_2.$$

Therefore, assuming the gradient of $\phi$ is defined, and the limit exists, it must satisfy:

$$\left|\frac{\partial\phi(\Theta)_{\tau\mu}}{\partial\Theta_{ij}}\right| \leq \sup_{\Delta\Theta_{ij}\neq 0} \frac{\sqrt{h}\|\phi(\Theta) - \phi(\Theta + \Delta\Theta)\|_2}{\|\Delta\Theta\|_2} \leq k\sqrt{h} \leq 4\sqrt{h}\log_2 h, \tag{36}$$

by the derivations presented for VPRNN (Taylor-Melanson et al., 2021). This completes the derivation of the $M_\phi$ bound for the VPRNN trivialization mapping.

### A.4 JUSTIFICATION OF CODE IMPLEMENTATION

The public code implementation of the VPRNN layer we required for our experiments adds some additional permutations to the matrix decomposition of $\mathbf{U} = \phi(\Theta)$. Regardless of how many it adds, we can define the mapping $\tilde{\phi}$ of the implementation as $\tilde{\phi} = \tilde{\mathbf{Q}}_1 \phi \tilde{\mathbf{Q}}_2$ for permutation matrices $\tilde{\mathbf{Q}}_1, \tilde{\mathbf{Q}}_2$, where $\phi$ is a typical VPRNN mapping using suitable permutation matrices. This definition is due to the fact that the product of two permutation matrices is also a permutation matrix. Using the spectral norm $\|\cdot\|_2$, we get the following inequality since $\tilde{\mathbf{Q}}_1, \tilde{\mathbf{Q}}_2$ are orthogonal:

$$\left\|\tilde{\phi}(\Theta + \Delta\Theta) - \tilde{\phi}(\Theta)\right\|_2 = \left\|\tilde{\mathbf{Q}}_1 \phi(\Theta + \Delta\Theta)\tilde{\mathbf{Q}}_2 - \tilde{\mathbf{Q}}_1 \phi(\Theta)\tilde{\mathbf{Q}}_2\right\|_2$$

$$\leq \left\|\phi(\Theta + \Delta\Theta)\tilde{\mathbf{Q}}_2 - \phi(\Theta)\tilde{\mathbf{Q}}_2\right\|_2$$

$$\leq \|\phi(\Theta + \Delta\Theta) - \phi(\Theta)\|_2$$

$$\leq k \|\Delta\Theta\|_2.$$

So, the result required for Corollary 1 holds for the implementation provided and the extra permutations can be ignored.

