# OpenReview forum: "SGORNN: Combining Scalar Gates and Orthogonal Constraints in Recurrent Networks"
_ICLR.cc/2022/Conference — ICLR 2022 Submitted_

### Official Review · Reviewer_aPoP · 2021-10-25

**Correctness:** 4
**Technical Novelty And Significance:** 3
**Empirical Novelty And Significance:** 2
**Recommendation:** 5
**Confidence:** 4

**Main Review:**

The idea of combining residual connection (scalar gates) and unitary RNN (orthogonal recurrent matrices)  is simple, well-documented, well-explained and well-demonstrated. In fact, this is one of these very nice simple ideas that can make the lives of many researchers way easier when playing with RNN (**only**). Indeed, the implementation of this solution (such as given in supplementary materials) is pretty straightforward.

Nevertheless, I am concerned by the impact and scalability of this approach. The given experiments are two simple toy-tasks (one would have been sufficient to validate the model conceptually) and a medium one with language modelling with the Penn TreeBank dataset. Unfortunately, and appart from the first toy-task, the experiments fail, imho, at giving insights on how well this model will behave in practice.

The major improvement of this work comes from the possibility to avoid exploding gradients and marginally improve the generalisation capabilities of the model. Hence, we are expecting experiments that demonstrate these points in realistic conditions. Exploding gradients on the addition task is not the same gradient-explosion-level that one may observe with large scale datasets. How is SGORNN behaving with complex tasks that are known to cause gradient explosion, like speech processing or very-long sequence processing with large input vectors and large architectures ? How is SGORNN performing against existing RNN on well-known sequence modelling task ? Penn TreeBank with a perplexity of 130 is quite high. We do not require a SoTA results, this is out of the scope of such a paper, however, it remains that a new researcher can not have a clear idea of if this idea will work and scale well in his/her own complex scenario. In practice, most of the time, we are not just doing additions. An other point along these lines is that Vanilla RNN aren't the must-go solution right now. Practically speaking, RNN in general (LSTM, GRU others) are even being left in favour of self-attention. Hence, to have real impact on the community, this work must be extended to architectures that are widely used and still suffer from gradient explosion (GRU).

**Strengths**
- Well written
- A nice, simple and effective (under certain conditions) idea.

**Weaknesses**
- Real impact on the community quite limited as only vanilla RNN are concerned by the study.
- Experiments fail at proving the points addressed by the proposed method (at scale and with realistic conditions).

**Minor remarks:**

Contrast for pictures are not well suited for black and white printers ! Impossible to distinguish the curves.
The conclusion isn't really a conclusion but more of discussion + conclusion. It would be better to split it in two different parts.



**Summary Of The Paper:**

This work proposes to approach and partially solve (under certain constraints) the exploding gradient problem of RNN. In practice, the authors combined two existing methods (residual connection + unitary RNNs) in a single novel model. Demonstrations are given for the exploding issue and for the motivation of parameter constraints. The new model, called SGORNN, is then tested on two toy tasks and on a largish one (language modelling). SGORNN outperform models that are based either on residual connection or unitary recurrent connections only.

**Summary Of The Review:**

This paper presents a nice, simple and well-motivated idea. However, it fails at demonstrating how impactful it could be to the community with experiments that are not supporting fully the claims at scale or in realistic conditions.

---

> ### Author Response · Authors · 2021-11-21
> **Rebuttal Comments for Reviewer aPoP**
>
> Thank you for your review. Please find our response below to your comments:
>
> >I am concerned by the impact and scalability of this approach. The given experiments are two simple toy-tasks (one
> would have been sufficient to validate the model conceptually) and a medium one with language modelling with the
> Penn TreeBank dataset. Unfortunately, and appart from the first toy-task, the experiments fail, imho, at giving
> insights on how well this model will behave in practice.
>
> Thank you for your feedback. On your point that we used two toy tasks, we respectfully disagree. The
> **second task, HAR-2, is not a toy task. It operates on real-world data recorded on smartphones and is taken
> from FastRNN for comparison.** The HAR-2 task is a practical task, necessary for research on deploying
> RNNs on embedded systems and smartphones.
>
> >Exploding gradients on the addition task is not the same gradient-explosion-level that one may observe with large
> scale datasets. How is SGORNN behaving with complex tasks that are known to cause gradient explosion, like
> speech processing or very-long sequence processing with large input vectors and large architectures ? How is
> SGORNN performing against existing RNN on well-known sequence modelling task ?
>
> Thank you for this feedback. **Such tasks as you mention will be the topic of future work.**
>
> >Penn TreeBank with a perplexity of 130 is quite high. We do not require a SoTA results, this is out of the scope of
> such a paper, however, it remains that a new researcher can not have a clear idea of if this idea will work and scale
> well in his/her own complex scenario. In practice, most of the time, we are not just doing additions.
>
> While we agree that perplexity value of 130 may be considered high, **we note that this is similar to that
> seen for FastRNN in the original publication (arXiv:1901.02358)**. In short, **the goal of the experiment was
> to compare with FastRNN, and our results show that SGORNN is comparable with FastRNN in terms of
> test perplexity on Penn TreeBank.**
>
> >to have real impact on the community, this work must be extended to architectures that are widely used and still
> suffer from gradient explosion (GRU).
>
> We note that **the combination of orthogonal constraints with GRU has been studied (arXiv:1706.02761)** as mentioned by another reviewer, but other possibilities include combining SGORNN with FastGRNN (rather than FastRNN), which may
> be the topic of future work. This initial work did not address combinations with gated RNNs due to the
> goal of reducing the model’s parameter space size. Please note that **we have now added LSTM as a
> baseline to our experiments.**
>
> >Contrast for pictures are not well suited for black and white printers ! Impossible to distinguish the curves.
>
> Thank you for bringing this to our attention, we agree. **All pictures are now created using B&W line
> styles.**

---

> > ### Comment · Reviewer_aPoP · 2021-11-24
> > **Thanks for your answers**
> >
> > First, I would like to acknowledge the work done by the authors on this rebuttal. Second, I would like to rectify one mention I did about HAR-2. I did not want to say that HAR-2 was unrealistic. It is, however, a small dataset that doesn't give much insight about the scaling capabilities of SGORNN. I also respectfully disagree with the statement that the scaling task should be left for future work. The kind of problems that the author's model seems to solve are typically the ones that we observe when dealing with these tasks -> Speech Recognition, Speech Translation, Long-term forecasting etc. Having just 1 task among those would boost the validity (and impact of this work).
> >
> > Unfortunately, to me, this work still fails at convincing the reader about the transferability of the technique. I will keep an eye on the discussion with the other reviewers, but for now, my score will remain the same.

---

### Official Review · Reviewer_W7nS · 2021-10-27

**Correctness:** 3
**Technical Novelty And Significance:** 1
**Empirical Novelty And Significance:** 1
**Recommendation:** 3
**Confidence:** 4

**Main Review:**

1. The proposed model has very limited novelty. It simply put the orthogonal constraint on a standard FastRNN. Combining orthogonal transition matrices with other mechanisms in RNNs has been studied in the literature (arXiv:1706.02761), but the authors failed to mention any of them.

2. The orthogonal constraint method used in the paper is from VPRNN (Taylor-Melanson et al., 2021), which seems problematic. It is not proven to cover the entire orthogonal space and appears to have heavily redundant operations. Since the paper's main contribution is to combine orthogonal constraint with FastRNN (Kusupati et al., 2019), it is necessary to show that the combination works for ANY orthogonal parametrization methods (arXiv:1511.06464, arXiv:1612.00188, arXiv:1612.05231, arXiv:1901.08428).

3. The theory part is very impressive. It starts with FastRNN's original theorems and derives a convincing conclusion based on the orthogonal condition.

4. Experiments only compared the proposed model against FastRNN and VPRNN. It's more like a simple ablation section to show both components in the model are functioning. However, this is far from showing the model is valid. It failed to compare to any modern orthogonal RNN architectures or modern gated RNNs. For example, the Penn Treebank perplexity results are all far from basic LSTM/GRU performance. Therefore, the experiments cannot justify the method.


**Summary Of The Paper:**

This paper proposed a novel recurrent network architecture called Scalar Gated Orthogonal Recurrent Neural Networks (SGORNN). The model is a combination of (1) orthogonal transition matrix and (2) fixed residual connection as in FastRNN (Kusupati et al., 2019).
The authors theoretically studied the generation bound and gradient exploding condition of the model.
Lastly, experiment results show that the proposed model outperforms standard FastRNN and a version of vanilla orthogonal RNN on three tasks: (1) synthetic adding problem, (2) HAR-2 classification, (3) Penn Treebank word-level language modeling.

**Summary Of The Review:**

The method proposed in the paper has limited novelty. The theory section in the paper has a meaningful contribution.
The experiments are not convincing.
Overall, the paper is not appropriate to publish on ICLR.

---

> ### Author Response · Authors · 2021-11-21
> **Rebuttal Comments for Reviewer W7nS**
>
> Thank you for your review. We have attempted to address your concerns below:
>
> >The proposed model has very limited novelty. It simply put the orthogonal constraint on a standard FastRNN.
> Combining orthogonal transition matrices with other mechanisms in RNNs has been studied in the literature (arXiv:1706.02761), but the authors failed to mention any of them.
>
> It should be noted that **GORU (arXiv:1706.02761) also simply adds an orthogonal constraint to another RNN architecture (GRU), so the novelty of the concepts themselves are very similar in our view**. However, we appreciate you noting our failure to mention this work and **we have now added mention to GORU in our paper**. Your final phrasing “any of them” makes us wonder if there is any additional works you think we may have failed to cite, and **we would appreciate a mention of any additional related orthogonal-combination architectures in your reply**.
>
> >The orthogonal constraint method used in the paper is from VPRNN (Taylor-Melanson et al., 2021), which seems
> problematic. It is not proven to cover the entire orthogonal space and appears to have heavily redundant operations.
> Since the paper's main contribution is to combine orthogonal constraint with FastRNN (Kusupati et al., 2019), it is
> necessary to show that the combination works for ANY orthogonal parametrization methods
>
> We briefly compare here the VPRNN architecture to the EURNN architecture (arXiv:1612.05231) used
> by GORU (arXiv:1706.02761). Both architectures rely on 2x2 rotation matrices and permutations to construct orthogonal matrices,
> though VPRNN does not require use of a complex exponential, **which leads us to believe our chosen parametrization is not more redundant than the one used in the work you suggest we cite.**
>
> Regarding covering the orthogonal space, this is a valid criticism of the VPRNN parametrization.
> However, it should be noted that **covering the entire orthogonal space may not be necessary, as (Taylor-Melanson et al., 2021) provides experimental results surpassing those previously recorded for EURNN**.
>
> Finally, regarding showing our architecture works for any orthogonal parametrization, **the use of
> additional parametrizations will be the topic of future work** (as mentioned in our final section). It is worth
> noting, though, that **the GORU architecture you suggest we cite (arXiv:1706.02761) uses only the
> EURNN parametrization in experiments**. For this reason, **we believe there is precedent for architectures
> similar to our own using only one parametrization within the experimental section.**
>
> >It failed to compare to any modern orthogonal RNN architectures or modern gated RNNs. For example, the Penn
> Treebank perplexity results are all far from basic LSTM/GRU performance. Therefore, the experiments cannot
> justify the method.
>
> Thank you for this criticism. **We have now added an LSTM baseline to our experiments**. We note that
> while LSTM has shown low perplexity on PTB (on the order of 80 PPL in (arXiv:1803.01271)), **such models had over 13M parameters, nearly three times the size of the models trained in our paper.** For this reason, you will notice that our LSTM baseline still has on the order of 130 PPL, as it has the same number of hidden units as other architectures trained. We also note that **although you see VPRNN
> as problematic as a parametrization, it is in our view a modern orthogonal RNN baseline as it is a very recent publication with promising experimental results.**

---

> > ### Comment · Reviewer_W7nS · 2021-11-28
> > **Response to authors**
> >
> > I appreciate the detailed response from the authors. Most part of my concerns has been addressed.
> >
> > The discussion comparing the proposed model to previous hybrid orthogonal models are convincing. I agree with the authors that covering the entire orthogonal space is not necessary.
> >
> > However, I find the discussion on the LSTM baseline experiment is far from being acceptable. The authors bring an LSTM result with better performance by a larger model. This is not meaningful. The authors need to either train the model with the same size as the baseline LSTM or compare it to a smaller LSTM model.
> >
> > I think the paper has been improved. But considering the fatal problem in this PTB experiment and the overall novelty of the paper, my score will remain the same.

---

### Official Review · Reviewer_NMkH · 2021-11-02

**Correctness:** 3
**Technical Novelty And Significance:** 3
**Empirical Novelty And Significance:** 3
**Recommendation:** 6
**Confidence:** 4

**Main Review:**

I thought this was a good submission. The model architecture is relatively novel yet simple to implement, the bounds are easier to satisfy compared to FastRNN or VPRNN, but most importantly the model's performance is far superior to the baselines. The experiments were well done in that a relatively large hyperparameter sweep was done in order to induce gradient explosions.

The one issue I have is that perhaps slightly less relevant, harder baselines should have been chosen so as to show how much work there is to do to get a SGORNN-like architecture to converge to performance in more typical models.

All in all, I think this paper provides an interesting theoretical contribution that the community would appreciate.

**Summary Of The Paper:**

Authors demonstrate a new RNN-based architecture that avoids exploding gradients via careful selection of primitives and hyperparameters. The authors motivate their selections with gradient magnitude bounds and show proofs for them. Finally, the authors demonstrate their model's superior performance against several RNN-based baselines directly relevant to their architecture.

**Summary Of The Review:**

An interesting model with nice properties that does relatively well empirically, but more baselines (eg, orthogonal RNNs) should be included before the paper is accepted.

---

> ### Author Response · Authors · 2021-11-21
> **Rebuttal Comments for Reviewer NMkH**
>
> Thank you very much for your review. Please find our rebuttal below:
>
> >The one issue I have is that perhaps slightly less relevant, harder baselines should have been chosen so as to show
> how much work there is to do to get a SGORNN-like architecture to converge to performance in more typical
> models.
>
> Thank you for this feedback. **We have added an LSTM baseline to all three experiments** and hope this
> improves the quality of the paper in your opinion. Further, we wish to note that **VPRNN provides an
> orthogonal RNN baseline in our paper**. Additional orthogonal RNN baselines will be considered in future
> work.

---

### Official Review · Reviewer_815o · 2021-11-03

**Correctness:** 4
**Technical Novelty And Significance:** 1
**Empirical Novelty And Significance:** 1
**Recommendation:** 3
**Confidence:** 3

**Main Review:**

The paper is clearly written and easy to follow. However, the proposed method seems to be a simple extension to the FastRNN model, and has limited novelty. The only addition to the FastRNN model is the use of orthogonal weight matrices, which is also a well-established line of research in the field. The experimental results could also be improved by adding more baseline methods, including RNNs with orthogonal weight matrices, as listed in the introduction section.

**Summary Of The Paper:**

In this paper, the authors propose an RNN architecture named SGORNN. The high-level idea is to add a residual connection between hidden states, and to parametrize the weight matrix by the product of a series of rotation and permutation matrices so that the weight matrix is orthogonal. The proposed model can be seen as an extension of the FastRNN model, the theoretical analyses of which also apply as a result. The proposed model is evaluated on a few tasks including the addition problem, HAR-2 classification, and PTB.

**Summary Of The Review:**

The proposed method seems to be a simple extension of an existing method (FastRNN) and thus has limited novelty.

---

> ### Author Response · Authors · 2021-11-21
> **Rebuttal Comments for Reviewer 815o**
>
> Thank you for your review. We present the following response and changes to our paper based on your
> comments:
>
> >the proposed method seems to be a simple extension to the FastRNN model, and has limited novelty. The only  addition to the FastRNN model is the use of orthogonal weight matrices, which is also a well-established line of research in the field.
>
> While we agree that our model is an extension of FastRNN, we believe that it provides novel improvement for the following reasons. First, **the generalization of SGORNN can be enforced at training time simply by ensuring $\beta \leq 1-2\alpha$**, while the generalization of FastRNN cannot be enforced during training in general. Second, **our model alleviates exponential gradients by adding the orthogonal weight matrix (a property that is not present in FastRNN) while decreasing the number of parameters needed to achieve the standard accuracy of FastRNN** by deploying the VPRNN parametrization.
>
> >The experimental results could also be improved by adding more baseline methods, including RNNs with
> orthogonal weight matrices, as listed in the introduction section.
>
> We agree with this criticism, and with this in mind **we have added LSTM as a baseline to all experiments**. We will note the
> use of other orthogonal RNNs (other than VPRNN, which is **an included orthogonal baseline**) as baselines
> for future work.

---

> > ### Comment · Reviewer_815o · 2021-11-30
> > **Response**
> >
> > I appreciate the detailed response from the authors. However, I am still not convinced that the level of novelty meets the acceptance threshold. Therefore, I would like to keep my score.

---

### Decision · Program_Chairs · 2022-01-20

**Decision:**

Reject

**Comment:**

This paper considers the exploding gradient problem in RNNs. The proposed network SGORNN can be seen as an extension to the FastRNN model by adding orthogonal weight matrices.

I recommend rejection for this paper mainly for two reasons.

First, as mentioned in the review of Reviewer 815o and Reviewer W7nS, adding orthogonal constraints into FastRNN should not be considered as a significant technical contribution.

Second, more importantly, the experiments of the paper are not that convincing. All reviewers raise concerns about this issue. I also do not see the point of comparing the proposed model with a baseline LSTM model of much larger parameter size. I can’t think of a reason to do so. Also I think the small datasets will not give you a lot of meaningful insights in comparing the models – PTB for example, is a rather small dataset for language modeling and the results presented there are far from well. The numbers look really bad, reflecting the quality of how these experiments are done ( https://arxiv.org/pdf/1707.05589.pdf ).